# Enhancement of Antioxidant and Anti-Glycation Properties of Beeswax Alcohol in Reconstituted High-Density Lipoprotein: Safeguarding against Carboxymethyllysine Toxicity in Zebrafish

**DOI:** 10.3390/antiox12122116

**Published:** 2023-12-14

**Authors:** Kyung-Hyun Cho, Seung-Hee Baek, Hyo-Seon Nam, Ashutosh Bahuguna

**Affiliations:** Raydel Research Institute, Medical Innovation Complex, Daegu 41061, Republic of Korea; shbaek@raydel.co.kr (S.-H.B.); sun91120@raydel.co.kr (H.-S.N.); ashubahuguna@raydel.co.kr (A.B.)

**Keywords:** beeswax alcohol (BWA), high-density lipoproteins (HDL), reconstituted HDL (rHDL), low-density lipoproteins, carboxymethyllysine (CML), zebrafish, embryo

## Abstract

The antioxidant and anti-inflammatory abilities of beeswax alcohol (BWA) are well reported in animal and human clinical studies, with a significant decrease in malondialdehyde (MDA) in the blood, reduced liver steatosis, and decreased insulin. However, there has been insufficient information to explain BWAs in vitro antioxidant and anti-inflammatory activity owing to its limited solubility in an aqueous buffer system. Herein, three distinct reconstituted high-density lipoproteins (rHDL) were prepared with palmitoyloleoyl phosphatidylcholine (POPC), cholesterol, apolipoprotein A-I (apoA-I), and BWA at molar ratios of 95:5:1:0 (rHDL-0), 95:5:1:0.5 (rHDL-0.5), and 95:5:1:1 (rHDL-1) and examined for antioxidant and anti-glycation effects. A rHDL containing BWA, precisely rHDL-1, displayed a remarkable anti-glycation effect against fructose (final 250 mM), induced glycation of HDL, and prevented proteolytic degradation of apoA-I. Also, BWA incorporated rHDL-0.5, and rHDL-1 displayed substantial antioxidant activity by inhibiting cupric ion-mediated low-density lipoprotein (LDL) oxidation. In contrast to rHDL-0, a 20 and 22% enhancement in ferric ion reduction ability (FRA) and paraoxonase (PON) activity was observed in HDL treated with rHDL-1, signifying the effect of BWA on the antioxidant activity enhancement of HDL. rHDL-1 efficiently inhibits *N*^ε^-carboxylmethyllysine (CML)-induced reactive oxygen species (ROS) generation and apoptosis in zebrafish embryos, consequently improving embryo survivability and developmental deformities impaired by the CML. The dermal application of rHDL-1 to the CML-impaired cutaneous wound of the adult zebrafish inhibited ROS production and displayed potent wound-healing activity. Conclusively, incorporating BWA in rHDL significantly enhanced the anti-glycation and antioxidant activities in rHDL via more stabilization of apoA-I with a larger particle size. The rHDL containing BWA facilitated the inherent antioxidant ability of HDL to suppress the CML-induced toxicities in zebrafish embryos and ameliorate CML-aggravated chronic wounds in adult zebrafish.

## 1. Introduction

Many hydrophilic antioxidants, such as vitamin C, allicin, and melatonin, have been developed and marketed to react with oxidants in the cell cytoplasm and the blood plasma [1]. On the other hand, water-soluble antioxidants have limitations in inhibiting lipid peroxidation in the cell membranes because of their low interaction with the hydrophobic interface [2]. The development of lipid-soluble antioxidants is more advantageous to prevent oxidative damage to the cell membrane in a nonpolar environment [3]. Many chronic aging-related diseases are frequently associated with oxidative damage in cell membranes by reactive oxygen and nitrogen species [4,5], which have amphipathic properties between water and lipid interfaces. On the other hand, lipid-soluble antioxidants, such as carotenoids, vitamin E, coenzyme Q_10_ (CoQ_10_), docosahexaenoic acid (DHA), and beeswax alcohol (BWA), are well-known nutrients with the potential to protect the cell membrane and treat chronic aging-related diseases, such as cardiovascular disease (CVD), diabetes, and neurodegenerative diseases [6]. These compounds may prevent the cell membranes from being damaged by free radicals by readily scavenging peroxyl radicals, preventing lipid and protein oxidation in the amphipathic interface of the plasma membrane in brain cells, neurons, and glia [7]. Nevertheless, in vitro evidence has not been sufficiently accumulated to prove the mechanistic insight of the antioxidants due to their low solubility in water.

Beeswax alcohol (BWA), a beeswax-extracted substance, comprises a blend of six main aliphatic alcohols (of carbon chains 24, 26, 28, 30, 32, and 34) that demonstrate antioxidant, anti-platelet, cholesterol-lowering, and gastroprotective properties [8,9]. Notably, supplementing with BWA enhanced the production and quantity of gastric mucus in rat models with ethanol-induced ulcers and mitigated inflammation in cases of antigen-induced arthritis [10,11]. BWA is acknowledged for safeguarding the body against oxidative stress, promoting a healthy stomach by protecting the gastric mucosa, and contributing to maintaining joint health. With these attributes, the Ministry of Food and Drug Safety in Korea has officially recognized BWA as a functional food ingredient [12]. Oral acute treatment with BWA (5–100 mg/kg) reduces gastric ulceration and malondialdehyde (MDA) formation, a marker of lipid peroxidation, in the gastric mucosa of rats with indomethacin or ischemia reperfusion-induced ulcers [13]. The administration of BWA, 25–200 mg/kg, into rats by oral gavage revealed potent anti-inflammatory activity with a significant reduction of leukotriene B_4_ (LTB_4_) against carrageenan-induced pleuritic inflammation [14].

An in vitro assay of the BWA has been a limitation in evaluating its antioxidant capacity because of its extreme insolubility in the physiological aqueous buffer system. It is hard to select an isotonic aqueous buffer to solubilize BWA for enzyme assays, cell-based assays, and animal experiments. To obtain higher concentrations in in vitro experiments, the BWA has been frequently solubilized in detergent solutions, such as Tween20, which have limitations due to adverse effects at higher concentrations. The hurdle was overcome by incorporating BWA into reconstituted high-density lipoproteins (rHDL) with apoA-I to evaluate the antioxidant capacities of BWA in the physiological buffer system, according to previous reports [15,16]. The current study compared the in vitro effects of rHDL containing BWA (BWA-rHDL) and encapsulation of BWA into rHDL particles on the anti-glycation, antioxidant, and anti-inflammatory activities. The anti-glycation activity of BWA-rHDL was assessed through examination of fructose or carboxymethyllysine (CML)-induced glycation of HDL, a process linked to inflammation and neurotoxicity [17]. Elevated serum *N*^ε^-carboxymethyllysine (CML) levels were found to be associated with the worsening of atherosclerosis, attributed to lipoprotein modifications and heightened susceptibility of low-density lipoproteins (LDL) to oxidation [18]. To elevate the antioxidant properties of BWA, zebrafish embryos were employed, and their developmental speed, swimming ability, and survivability were examined after the injection of rHDL-BWA in the presence of CML. Zebrafish embryos, akin to mammals, possess well-developed innate and acquired immune systems [19]. The use of zebrafish embryos is advantageous due to their external development and optical transparency through their developmental stages. These attributes make zebrafish and their embryos a suitable animal model for various studies, such as the screening of antioxidants [20], wound healing [21], and tissue regeneration [22].

This research aimed to access the physicochemical characteristics of BWA-rHDL, focusing on particle size, morphology, and electromobility in relation to varying molar ratios of BWA. The structural and functional correlations of each rHDL were examined, along with their antioxidant, anti-glycation, and anti-inflammatory activities in vitro and in vivo, utilizing zebrafish embryos and adults subjected to acute hyperinflammation. 

## 2. Materials and Methods

### 2.1. Materials

*N*^ε^-carboxylmethyllysine (CML, CAS-No. 941689-36-7, Cat#14580) with at least >97% purity (by thin layer chromatography) with water (<12.0%) was purchased from Sigma-Aldrich (St. Louis, MO, USA). Palmitoyloleoylphosphatidylcholine (POPC, #850457) was provided by Avanti Polar Lipids (Alabaster, AL, USA). Sodium cholate (#1254) and fructose (Cat #F0127) were purchased from Sigma-Aldrich (St. Louis, MO, USA). Purified beeswax alcohol (BWA) with a purity exceeding 90% was sourced from the National Center for Scientific Research (CNIC) in Havana, Cuba, through Raydel Pty, Ltd. (Thornleigh, NSW, Australia). The BWA material consisted of six high-molecular-weight alcohols derived from beeswax (obtained from *Apis melliferous*, L.) with the following compositions: octacosanol (6–15%), hexacosanol (7–20%), octacosanol (12–20%), triacontanol (25–35%), dotriacontanol (18–25%), and tetratriacontanol (≤7.5%) with a purity of 85%) as shown in Appendix A [23].

### 2.2. Isolation of Human Lipoproteins by Ultracentrifugation

The various fractions of serum lipoproteins, specifically LDL (1.019 < density (d) < 1.063) and HDL (1.063 < d < 1.225), were extracted from blood samples obtained from humans with an average age of 23 ± 2 years. The participants voluntarily donated the blood following a 14 h fasting period, and the collection process adhered to the Helsinki guidelines. The Institutional Review Board of the Korea National Institute for Bioethics Policy (KoNIBP) approved this study with the authorization number P01-202109-31-009, supported by Korea’s Ministry of Health Care and Welfare (MOHW). Different lipoprotein fractions from the blood were segregated by density gradient ultracentrifugation, where different density zones were prepared using NaCl and NaBr according to the standard method [24]. Briefly, 10 mL of the blood sample was centrifuged at 4000× *g* for 20 min to obtain the serum. The serum (3 mL) was mixed with a density gradient mixture prepared by NaCl (1.019 < d < 1.063) and NaBr (1.063 < d < 1.225). The content was ultracentrifuges (Himac NX, Hitachi, Tokyo, Japan, equipped with a fixed-angle rotor P50AT4-0124) at 100,000× *g* for 24 h at 10 °C. The separated zones of LDL (with density 1.019 < d < 1.063) and HDL (with density 1.063 < d < 1.225) were recovered. LDL and HDL were individually subjected to dialysis against Tris-buffered saline [TBS; composed of 10 mM Tris-HCl, 140 mM NaCl, and 5 mM ethidium-diamine tetraacetic acid (EDTA), pH 8.0] to eliminate any remaining NaCl and NaBr traces. The final yields of LDL and HDL were 4 mg (with a protein concentration of 1 mg/mL) and 12 mg (with a protein concentration of 1 mg/mL), respectively.

### 2.3. Human apoA-I Purification

ApoA-I was purified from HDL using organic solvent extraction as described by Brewer et al. [25], with a slight modification. Briefly, the isolated HDL (~5 mg) was delipidated by blending it in 1 mL of organic solvent [chloroform: methanol (2:1, vol:vol)]. The delipidated apoA-I was purified by fast protein liquid chromatography using an AKTA purifier system (GE Healthcare, Uppsala, Sweden) equipped with a Superose 6 10/300 GL column (GE Healthcare). The apoA-I was eluted by 10 mM Tris-HCl/140 mM NaCl (pH 8.0). The protein purity of the isolated apoA-I (95%) was confirmed by SDS-PAGE analysis.

### 2.4. Oxidation of LDL in the Presence of Lipid-Free BWA

Oxidized LDL (oxLDL) was generated by incubating the human LDL (1 mg/mL, 75 μL) with CuSO_4_ (final 10 μM) following 4 h of incubation at 37 °C. The effect of BWA on preventing LDL oxidation was evaluated by adding 10, 20, and 30 μM BWA to the reaction mixture (LDL + CuSO_4_). After 4 h incubation, the oxLDL and the LDL treated with BWA were filtered using a 0.22 μm filter and analyzed to measure the extent of lipid oxidation using a thiobarbituric acid reactive substances (TBARS) assay [26] with malondialdehyde (MDA, Sigma #63287) standard and oxidation of protein using 0.5% agarose gel. During the reaction, the amount of oxidized product in the mixture was quantified based on the conjugated diene level by continuous monitoring of absorbance at 234 nm for 60 min using a UV-Vis spectrophotometer (UV-2600i, Shimadzu, Kyoto, Japan).

### 2.5. Anti-Glycation Activity of Lipid-Free BWA

The glycation process involved incubating 2 mg/mL of purified HDL (400 μL) with 400 μM of CML (160 μL). Subsequently, BWA solution was added to achieve 0, 10, 20, and 30 μM BWA concentrations. The volume of the reaction mixture was adjusted to 800 μL using 0.2 M KH_2_PO_4_/0.02% NaN_3_ buffer (pH 7.4). The content was incubated for 144 h at 37 °C in the presence of 5% CO_2_. Finally, the apoA-I content in HDL was evaluated by 15% SDS-PAGE (5 μg HDL protein in each lane) following densitometric analysis. The extent of glycation was evaluated by a significant reduction in protein content within HDL (apoA-I). The separated apoA-I band intensity (BI) was assessed for comparison through band scanning on a Chemi-Doc^®^ XR (Bio-Rad, Hercules, CA, USA) employing Quantity One software (version 4.5.2) across three separate SDS-PAGE runs.

The assessment of advanced glycation reactions was conducted by examining fluorescence intensity at 370 nm (excitation) and 440 nm (emission), as outlined in the previous study [27]. Concurrently, the wavelengths of maximum fluorescence (WMF) of tryptophan (Trp) residues in apoA-I were determined using an FL6500 spectrofluorometer (Perkin-Elmer, Norwalk, CT, USA). In the same sample, the WMF of tryptophan (Trp) residues in apoA-I was ascertained from the uncorrected spectra employing an FL6500 spectrofluorometer, following the earlier described method [15]. A quartz cuvette with a path length of 1 cm was utilized, and samples were exited at 295 nm to avoid excitation of tyrosine fluorescence. Subsequently, emission spectra were recorded across a wavelength range of 305 to 400 nm.

### 2.6. Synthesis of Reconstituted HDL and Electrophoresis

Reconstituted HDL (rHDL) was synthesized through the sodium cholate dialysis technique [28]. The initial molar ratios employed were 95:5:1:0, 95:5:1:0.5, and 95:5:1:1 for POPC:cholesterol:apoA-I:BWA, respectively. For the first POPC, cholesterol and BWA were mixed in chloroform and methanol (2:1, vol: vol) as the amount mentioned in Table 1. The contents were vortexed and processed for drying at 37 °C under a gentle stream of nitrogen. The dried content was suspended in Tris-buffered saline (TBS, pH 8.0) with occasional agitation, followed by the addition of sodium cholate solution (30 mg/mL) and apoA-I solution (1 mg/mL) to prepare rHDL (0.7 mL). The prepared rHDL was dialyzed overnight against TBS to remove the sodium cholate. After synthesis, each rHDL (9 μg of apoA-I/lane) was electrophoresed in denatured conditions using 14% SDS-PAGE to compare the presence of apoA-I and other ingredients in rHDL. Also, the prepared rHDL (10 μg of apoA-I/lane) was electrophoresed in a 0.5% agarose gel under native conditions to compare the particle size. The separated band intensities (BI) in both SDS-PAGE and agarose gel were determined by band scanning on a Chemi-Doc^®^ XR (Bio-Rad) employing Quantity One software (version 4.5.2).

### 2.7. Transmission Electron Microscopic (TEM) Analysis of Reconstituted HDL

The TEM analysis was conducted to assess the size and structure of the synthesized rHDL as described previously [15]. For the TEM analysis, an equal volume of rHDL was mixed with 1% sodium phosphotungstate (PTA, pH 7.4). Subsequently, 5 μL of suspension was blotted into the filter paper and swiftly substituted with another 5 μL of 1% PTA. After a few seconds, the stained rHDL was blotted on the Formvar carbon-coated 300 mesh copper grid, followed by the visualization at 40,000× *g* under transmission electron microscopy (TEM, Hitachi, model HT-7800; Ibaraki, Japan) at an acceleration voltage of 80 kV.

### 2.8. LDL Oxidation in the Presence of rHDL

The antioxidant capacity of each rHDL was tested to prevent LDL oxidation mediated by Cu^2+^ (final 10 μM). The LDL (2 mg/mL) was mixed with CuSO_4_ (final 10 μM) in the presence of different rHDL (as mentioned in Table 1) following 4 h of incubation at 37 °C. The extent of oxidation was evaluated by relative electromobility using a 0.5% agarose gel [29]. The LDL treated with different rHDLs was processed for 0.5% agarose gel electrophoresis under non-denatured conditions. Electrophoresis was performed at 50 V for 1 h in Tris-acetate-EDTA buffer (pH 8.0). Apo-B in LDL was visualized by Coomassie brilliant blue staining (final concentration: 1.25%). Oxidized LDL migrated faster towards the bottom of the gel due to apo-B fragmentation and increased the negative charge. Also, the effect of different rHDL on LDL lipid peroxidation was evaluated by using a thiobarbituric acid reactive substances (TBARS) assay [26] with malondialdehyde (MDA, Sigma #63287) standard.

### 2.9. Anti-Glycation Activity of BWA in rHDL 

Glycation was conducted by incubating the purified HDL (2 mg/mL, 300 μL) with the fructose (final 250 mM, 150 μL); subsequently, rHDL (as mentioned in Table 1) was added. Finally, the volume reaction mixture was adjusted to 600 μL using 0.2 M KH_2_PO_4_/0.02% NaN_3_ buffer (pH 7.4). The content was incubated for 96 h at 37 °C in the presence of 5% CO_2_. Finally, the apoA-I content in HDL was evaluated by 15% SDS-PAGE (5 μg HDL protein in each lane) following densitometric analysis. The extent of glycation was evaluated by a severe decrease in the protein content in HDL (apoA-I). The separated apoA-I band intensity (BI) was assessed for comparison through band scanning on a Chemi-Doc^®^ XR (Bio-Rad) employing Quantity One software (version 4.5.2) across three independent SDS-PAGE runs. The fluorescent intensity (at 370 nm (excitation) and 440 nm (emission)) of the HDL (apoA-I) + Fructose treated with different rHDL was recorded to examine the extent of the advanced glycation reactions. [15,16].

### 2.10. Protein Determination

The protein content of purified HDL and LDL was obtained through ultracentrifugation, and the reconstituted HDL was analyzed using the Lowry assay according to a slight modification of the method reported by Markwell et al. [30] using Folin & Ciocalteu’s phenol reagent (F9252, Sigma-Aldrich, St. Louis, MO, USA). Bovine serum albumin (BSA) was used as a standard for calibration in the Lowry assay. Lipid-free apoA-I and other protein quantifications were carried out using the Quick Start^™^ Bradford Protein Assay Kit (Bio-Rad #5000201), with BSA as the standard.

### 2.11. Antioxidant Activities in the rHDL

The ferric ion-reducing ability (FRA) was assessed following the procedure outlined by Benzie and Strain [31]. In summary, FRA reagents were freshly prepared by mixing 20 mL of 0.2 M acetate buffer (pH 3.6), 2.5 mL of 10 mM 2,4,6-tripyridyl-S-triazine (Fluka Chemicals, Buchs, Switzerland), and 2.5 mL of 20 mM FeCl_3_∙6H_2_O. The antioxidant activities of each rHDL were estimated by measuring the increase in absorbance induced by the ferrous ions that were generated. Freshly prepared, the FRA reagent (300 μL) was mixed with each rHDL (100 μg of protein in total, 1 mL) as an antioxidant source. Using a spectrophotometer, the FRA was subsequently assessed by recording the absorbance at 593 nm at two-minute intervals over a 60 min duration at 25 °C. 

The evaluation of paraoxonase-1 (PON-1) activity towards paraoxon involved assessing the hydrolysis of paraoxon to *p*-nitrophenol and diethylphosphate, catalyzed by the enzyme [32]. Each reconstituted high-density lipoprotein (rHDL) sample (20 μL, 1 mg/mL) was equally diluted and then combined with 180 μL of paraoxon-ethyl (Signa Cat. No. D-9286), along with a buffer solution [90 mM tris-HCl/3.6 mM NaCl/2 mM CaCl_2_ (pH 8.5)]. The determination of PON-1 activity was conducted by measuring the initial velocity of *p*-nitrophenol production at 37 °C. This measurement was performed by assessing the absorbance at 415 nm using a microplate reader (Bio-Rad model 680; Bio-Rad, Hercules, CA, USA).

### 2.12. Zebrafish Maintenance

Zebrafish and their embryos were maintained using standard protocols [33] and in compliance with the Guide for the Care and Use of Laboratory Animals [34] approved by the Committee of Animal Care and Use of Raydel Research Institute (approval code RRI-20-003, Daegu, Republic of Korea). The zebrafish were housed in a temperature-controlled system tank at 28 °C and subjected to a 10:14 h light cycle. The fish were fed a regular diet of tetrabit granules (Tetrabit Gmbh D49304, Melle, Germany).

### 2.13. Anti-Inflammatory Activity of Lipid-Free BWA in Zebrafish

Zebrafish were subjected to acute inflammation through intraperitoneal injection of 250 μg carboxymethyllysine (CML) in 10 μL of PBS, corresponding to 3 mM CML (when adjusted for the average body weight of 300 mg). The zebrafish were randomly allocated into five groups, with each group consisting of 30 individuals (n = 30). In Group I, zebrafish received an injection of 10 μL PBS containing 250 μg CML. In comparison, zebrafish in Groups II and III were co-injected with 250 μg CML with 10 μL of BWA (final 100 mM) and vitamin C (final 100 mM), respectively. A 28-gauge needle was employed for injection, introduced into the abdominal region of zebrafish following their anesthesia with 0.1% 2-phenoxyethanol. The zebrafish swimming behavior and survivability assessment occurred 60 min after treatment, utilizing previously outlined parameters [17]. The primary criteria for evaluating swimming activity encompassed tail fin motion and the occurrence of body convulsions [17,35]. For plasma analysis, blood samples were obtained from distinct zebrafish groups. In a nutshell, 2 μL of blood from various zebrafish groups were promptly combined with 3 μL of PBS containing ethylenediaminetetraacetic acid (EDTA, final concentration 1 mM). Aspartate transaminase (AST) (AM-201) and alanine transaminase (ALT) (AM-103K) levels were determined using a commercial detection kit (Asan Pharmaceutical, Hwasung, Republic of Korea), following the manufacturer’s instructions.

### 2.14. Liver Histology

Liver tissue, obtained through surgical extraction from each experimental group, underwent preservation in 10% formalin for 24 h. Following alcohol dehydration, the tissue was embedded in paraffin, and 5 μm thick sections were treated with poly-L-lysine and stained with Hematoxylin and Eosin (H&E). Examination of the stained tissue sections for morphological changes was conducted using an optical microscope (Motic Microscopy PA53MET, Hong Kong, China). The Image J platform (http://rsb.info.nih.gov/ij/, accessed on 16 May 2023) was utilized to quantify the nucleus-stained area by converting the native H&E-stained nuclei to red intensity.

The quantification of IL-6 production in hepatic tissue was conducted through immunohistochemical staining, employing a previously established method [35]. Briefly, the 5 μm thick tissue section was subjected to a primary anti-IL-6 antibody (ab9324, Abcam, London, UK). Following overnight incubation at 4 °C, the tissue section was developed using Envision + system Kits (code 40001, Dako, Denmark) that included a horseradish peroxidase (HRP) conjugated-secondary antibody specific to the primary anti-IL-6 antibody.

### 2.15. Microinjection of CML and rHDL into Zebrafish Embryos

Zebrafish embryos at one-day post-fertilization (dpf) were microinjected individually using a pneumatic picopump (PV830; World Precision Instruments, Sarasota, FL, USA) equipped with a magnetic manipulator (MM33; Kantec, Bensenville, IL, USA) and a pulled microcapillary pipette-using device (PC-10; Narishigen, Tokyo, Japan). An injection of each rHDL alone (16 μg of apoA-I) or co-injection with CML (800 ng) was performed at the same location in the yolk to minimize bias, following a previously described method [16,17]. After the injection, the live embryos were observed under a stereomicroscope (Zeiss Stemi 305, Oberkochen, Germany) and photographed at 20× magnification using a ZEISS Axiocam 208 color (Jena, Germany). At 24 h post-injection, the chorion was removed, and each live embryo was compared to assess the developmental stage at a higher magnification of 50×.

### 2.16. Visulizing Oxidative Stress and, Apoptosis in the Embryo

After injecting CML with each rHDL, the levels of reactive oxygen species (ROS) and the degree of cellular apoptosis in the embryos were visualized through dihydroethidium (DHE) staining and acridine orange (AO) staining, respectively, by established procedures [35]. Fluorescence observations were employed to capture ROS images (Excitation = 585 nm and Emission = 615 nm), as outlined previously [36]. The assessment of cellular apoptosis across the various groups was conducted using acridine orange (AO) staining and fluorescence observations (Excitation = 505 nm, Emission = 535 nm), as reported elsewhere [37], using a Nikon Eclipse TE2000 microscope (Tokyo, Japan).

### 2.17. Cutaneous Wound Formation

A cutaneous wound was produced by anesthetizing 16-week-old zebrafish with 0.1% of 2-phenoxyethanol and removing the surface scale. A 2 mm-diameter cutaneous wound was produced using a biopsy punch (Kai Industries Co., Ltd., Oyana, Japan) at the left flank of the anal and dorsal fin regions of the zebrafish. The wounded zebrafish were segregated randomly into five groups (n = 15 in each group). The wound area of the zebrafish in Groups I and II was treated with 1 µL of TBS and 1 µL of 25 mg/mL CML (final 25 µg), respectively. The wound areas of the zebrafish in Groups III, IV, and V were treated with 1 µL of rHDL-0, rHDL-0.5, and rHDL-1 along with CML (final 25 µg), respectively. Three minutes post-treatment, the zebrafish were transferred into their respective tanks, maintained at a 28 °C water temperature.

### 2.18. Visual Observation of Wound Healing

The wounded area of the zebrafish was persistently monitored at 0, 2, 4, 6, 24, and 48 h after being stained with methylene blue as an earlier adopted method [21]. Briefly, at the respective time point, zebrafish were anesthetized by drenching into 0.1% of 2-phenoxyethanol and the subsequent addition of methylene blue (0.1% *w*/*v*, final 2 µL) at the wounded site. After one minute, the stained area was washed three times with water and visualized under a microscope. The wound area (blue-stained) was measured at different time points (2–48 h). Wound healing was calculated by comparing the wound area (mm^2^) at different time points with the wound area of 0 h, employing the Image J software (version 1.53).

### 2.19. Statistical Analysis

The outcomes are displayed as mean ± SD based on a minimum of three separate experiments. The SPSS software (version 29.0; SPSS, Inc., Chicago, IL, USA) was utilized for the statistical analysis. For the in vitro studies, the effects of each rHDL treatment were compared using an independent *t*-test. A *p*-value < 0.05 was considered significant.

## 3. Results

### 3.1. Antioxidant Ability of BWA against LDL Oxidation

The cupric ion (final 10 μM) treatment of native LDL (lane N) caused the production of more oxidized LDL (lane O), resulting in faster electromobility to migrate to the bottom of the agarose gel due to an increase in negative charge and apo-B fragmentation (Figure 1A). The co-treatment of vitamin C (vit C) resulted in adequate inhibition ability with slower electromobility and stronger band intensity (lane 3) than those of oxidized LDL. On the other hand, co-treatment of BWA (lanes 4–6) resulted in stronger antioxidant ability than vit C treatment to show the slowest electromobility and the strongest band intensity in a dose-dependent manner of BWA (lane 6).

A similar result on the LDL/apo-B electrophoretic mobility was noticed when the effect of BWA was compared with the lipid-soluble vitamin E (Appendix A), signifying the efficacy of BWA in preventing cupric ion-mediated LDL oxidation. The quantification of malondialdehyde (MDA) in the LDL mixture showed that the BWA treatment caused a significant decrease in MDA in a dose-dependent manner, up to 66% lower than oxidized LDL. In comparison, the vit C treatment showed only a 10% reduction at the final 30 μM treatment, as shown in Figure 1B. The treatment of BWA resulted in 62% lower MDA (*p* < 0.01) than the vit-C-treated LDL at the final 30 μM treatment in the presence of cupric ions and Tween20. In the presence of a 30 μM concentration, BWA treatment resulted in a 57% reduction in the MDA level compared to Vitamin E treatment, as illustrated in Appendix A. 

Concurrently, the assessment of conjugated dienes through absorbance measurements at 234 nm (A_234_) indicated that cupric ion treatment progressively elevated A_234_, reaching a peak of 173% at 60 min (relative to the initial value at 0 min, Appendix A). Conversely, the A_234_ of LDL without cupric ion remained stable over the 60 min incubation period. LDL treated with BWA (at 10 and 30 μM) demonstrated a significant protective effect against cupric ion-mediated LDL oxidation (Appendix A), as evidenced by negligible increases in A_234_ during the 60 min incubation compared with 0 min. In contrast, at concentrations of 10 and 30 μM, vitamin C exhibited less effectiveness in preventing cupric ion-mediated conjugated diene formation. At 10 and 30 μM concentrations, vitamin C-treated LDL showed elevation of A_234_ and reached 107% and 111%, respectively, at 60 min (compared to the initial value at 0 min) (Appendix A). These results collectively underscore BWAs substantially superior antioxidant capacity, surpassing both vitamin C and E against cupric ion-mediated LDL oxidation. This suggests the exceptional antioxidant efficacy of BWA in a hydrophobic environment, likely attributed to its lipophilic interaction with the LDL surface.

### 3.2. Anti-Glycation Activity of BWA against CML-Induced Modification of HDL

Lipid-free BWA exerted remarkable anti-glycation activity to suppress the production of yellowish fluorescence in a dose-dependent manner, with up to 11% and 13% inhibition of the fluorescence at 20 and 30 μM of BWA, respectively, at 144 h incubation (Figure 2A). Interestingly, WMF of HDL was red-shifted upon the CML treatment to 341 nm from 337 nm at the native state, suggesting that intrinsic Trp was exposed to the aqueous phase because of a disturbance of the tertiary structure by the glycation stress. On the other hand, the co-treatment of BWA caused a blue shift of WMF, especially around 338 nm at 30 μM of BWA, indicating the stabilization of apoA-I in the presence of BWA via the inhibition of glycation stress.

The electrophoresis of the HDL showed that the treatment of CML with HDL caused a remarkable degradation of the HDL band intensity with a slight up-shift in the band position (lane 0) compared to native HDL (lane N), as shown in Figure 2B. On the other hand, the co-treatment of BWA caused notable protection of apoA-I from CML-mediated proteolytic degradation in a dose-dependent manner: up to ~1.5-fold and 1.3-fold stronger band intensities were achieved by treatment of the final 30 μM of BWA at 72 h and 144 h, respectively. In addition, co-treatment of BWA resulted in a more distinct band intensity of apoA-I and shifted down the band position similar to that of native apoA-I (lane N). 

### 3.3. Anti-Inflammatory Activity of BWA against CML-Induced Acute Paralysis of Zebrafish

An intraperitoneal (IP) administration of CML resulted in acute paralysis observed at 60 min post-injection. In the CML-alone group, all zebrafish lie at the tank bottom with occasional quivering (Figure 3A). At 60 min, 13% of zebrafish in the CML-alone group exhibited a partial recovery of swimming ability, with 26% overall survivability (Figure 3B). However, the swimming pattern involved wobbling, seizures, and uncontrollable movements, as illustrated in Appendix A. Conversely, the group co-injected with BWA demonstrates the swiftest recovery of swimming ability. Approximately 47% of zebrafish in this group could swim again, displaying a more active and natural swimming pattern, resulting in a 57% survivability at 60 min post-injection, as depicted in Figure 3B and Appendix A. On the other hand, co-injection of Vit-C led to a slower recovery of swimming ability, with only around 10% of fish showing signs of swimming and a lower survivability rate of approximately 27% at 60 min post-injection, as shown in Figure 3B and Appendix A. This suggests that BWA treatment prevents paralysis and prevents zebrafish from CML-induced acute death.

### 3.4. Amelioration of Hepatic Damage

The CML+PBS group showed the highest level of AST and ALT levels around 951 ± 37 and 367 ± 33 IU/L (Figure 4), respectively, while the CML + BWA group showed the lowest levels of AST and ALT around 215 ± 13 and 107 ± 5 IU/L, respectively. However, the CML + vit C group showed higher levels of AST and ALT around 544 and 366 ± IU/L, respectively, than those of the CML + BWA group. 

### 3.5. Immunohistochemistry for Interleukin (IL)-6

As depicted in the images in Figure 5A, the immunohistochemical analysis of IL-6 in hepatic tissue demonstrated that the CML+PBS group exhibited the most extensive stained area, approximately 31.5%. In contrast, the CML + BWA group displayed the least IL-6-stained area, accounting for only 1.4%. Notably, the CML + Vit-C group revealed an IL-6-stained area of around 13.5%, demonstrating a significant 60% reduction (*p* < 0.001) compared to the CML + PBS group. Furthermore, this value was 8.9 times higher than observed in the CML + BWA group. These findings suggest that BWA exerts robust anti-inflammatory effects.

### 3.6. Synthesis of Reconstituted HDL Containing BWA

BWA exhibited sufficient binding ability to phospholipid (PL) and apoA-I to form rHDL (Table 1), with a molar ratio of 95:5:1:0 (rHDL-0), 95:5:1:0.5 (rHDL-0.5), and 95:5:1:1 (rHDL-1) for POPC:FC:apoA-I:BWA; the particle size of rHDL-1 was increased significantly, 15% higher than rHDL-0. The rHDL-0 and rHDL-0.5 showed a similar particle size of approximately 61–65 nm of diameter with the same wavelength maximum fluorescence (WMF) of 330.9–331.0 nm, indicating that the incorporation of a low molar ratio of BWA did not significantly change the rHDL structure and intrinsic Trp108 movement of the amphipathic helix in apoA-I. Upon lipid-free apoA-I binding with cholesterol and phospholipid in the native state, the WMF of apoA-I was 5.4 nm blue-shifted from 336.3 nm to 330.9 nm of WMF, indicating that the intrinsic Trp108 in apoA-I was moved to a more nonpolar phase. On the other hand, incorporating BWA at a high molar ratio resulted in a slightly larger redshift of WMF around 331.4 nm, indicating that Trp108 was more exposed and moved to the polar phase. After synthesis, each rHDL contained apoA-I (2 mg/mL) and rHDL-0.5, and rHDL-1 contained 12.5 μg/mL and 25 μg/mL of BWA, respectively.

### 3.7. Electrophoretic Profiles of rHDL Containing BWA

As shown in Figure 6A, the synthesis of rHDL was well carried out: the apoA-I band was up-shifted slightly upon binding with phospholipid and cholesterol (lanes 2, 3, and 4) compared to lipid-free apoA-I (lane 1). The rHDL containing BWA showed a thicker band area and intensity in the bottom of the gel for debris of PL and BWA (PL+BWA), as the red arrow indicated: rHDL-0.5 and rHDL-1 showed 1.6- and 2.8-fold, respectively, higher band intensity than that of rHDL-0. The BWA exhibited sufficient binding ability with phospholipid and apoA-I, with 17 and 53% more band intensity of apoA-I (lanes 3 and 4) than rHDL alone (lane 2), as shown in Figure 6A, indicating that more BWA binding is associated with more protection of apoA-I during the rHDL synthesis.

As shown in Figure 6B, under the native state, the three rHDL showed similar electromobility in 0.6% agarose, indicating that the three-dimensional structure of apoA-I and the electric charge distribution of rHDL particles were similar in all rHDL. On the other hand, rHDL-1 showed the strongest band intensity of apoA-I, while rHDL-0 showed the weakest band intensity. These results suggest that the greater incorporation of BWA in rHDL resulted in the stabilization of apoA-I and rHDL via the putatively strong affinity binding of BWA and apoA-I with the darker band intensity, as shown in the blue arrow (Figure 6B). 

### 3.8. Electron Microscopy Observations

Transmission electron microscopy (TEM) showed that rHDL-0 (photo a) had a typical discoidal rHDL shape and rouleaux morphology with a scattered pattern (Figure 7). In contrast, rHDL-1 (photo c) showed the most distinct disc particle shape with a long rouleaux morphology. In contrast, rHDL-0.5 showed a smaller particle shape and a short rouleaux morphology (photo c). The rHDL-0 and rHDL-0.5 showed similar particle sizes of 63–64 nm, but rHDL-1 showed the largest particle size with a homogeneous pattern around 70 ± 2 nm in diameter (inset graph D).

### 3.9. Anti-Glycation Activity of BWA in rHDL

Treating human HDL with fructose (250 mM) led to pronounced glycation, resulting in a seven-fold increase in yellowish fluorescence intensity (FI) compared to untreated HDL over a 90 h incubation under 5% CO_2_ (Figure 8A). Conversely, the application of rHDL-1 exhibited significant inhibition of fructose-induced HDL glycation, demonstrating a glycation extent up to 26% lower (*p* < 0.001) than glycation observed in HDL treated with fructose after 96 h of incubation. In contrast, other rHDLs (i.e., rHDL-0 and rHDL-0.5) also inhibited fructose-induced glycation, as evidenced by 5–16% less glycation than HDL treated with fructose. Electrophoretic analysis of each HDL sample indicated that native HDL (i.e., HDL alone) exhibited a distinct apoA-I band (28 kDa) at 0 h (lane 1) and 72 h (lane 2) in the absence of fructose treatment (Figure 8B). Glycated HDL (lane 3) showed a remarkably diminished apoA-I band; 90% had disappeared after the fructose treatment. In contrast, the rHDL-0 co-treatment resulted in a 6% larger apoA-I band intensity than fructose-treated HDL, suggesting that rHDL alone had a low protection effect of apoA-I against the proteolytic degradation of glycation. On the other hand, rHDL containing BWA-treated HDL showed a stronger band intensity of apoA-I with an increase in BWA content. Among rHDL, the rHDL-1-treated HDL (lane 6) showed the strongest band intensity: 28% more than rHDL-0. Hence, incorporating BWA in rHDL contributed to the protection of apoA-I from proteolytic degradation in a dose-dependent manner.

### 3.10. Antioxidant Ability of BWA in rHDL against LDL Oxidation

Native LDL exhibited the most pronounced band intensity with the slowest electromobility (lane N) (Figure 9A). Conversely, LDL treated with cupric ions (10 μM) displayed the least band intensity with the highest electromobility (lane O). On the other hand, co-treatment of rHDL (lane 1) resulted in a slight attenuation of electromobility with a larger increase in the LDL band intensity, suggesting that rHDL alone had low antioxidant activity. The co-treatment of rHDL-0.5 (lane 2) and rHDL-1 (lane 3) resulted in slower electromobility with stronger band intensity than those of rHDL-0 (lane 1), suggesting that incorporation of BWA in the rHDL exerted more antioxidant activity to inhibit LDL oxidation. More oxidized LDL migrated faster towards the bottom of the gel from the loading position at the top, showing a more smeared and weaker band intensity of LDL caused by fragmentation of apo-B and an increase in negative charge in LDL. 

Quantification of MDA in each LDL to determine the oxidation extent revealed the oxidized LDL (lane O) to have a 13-fold higher MDA level than native LDL (lane N), as shown in Figure 9B. On the other hand, the co-treatment of rHDL-0, rHDL-0.5, and rHDL-1 resulted in 9%, 24%, and 36% lower MDA than that of oxLDL, suggesting that incorporation of BWA in the rHDL exerted more antioxidant activity in a dose-dependent manner. 

### 3.11. Enhancement of the Antioxidant Ability of HDL by rHDL Containing BWA

The ferric ion reduction ability (FRA) of HDL (2 mg/mL) was enhanced by adding rHDL containing BWA, as shown in Figure 10A. HDL + rHDL-1 showed a 28% and 22% higher FRA than HDL alone and HDL + rHDL-0, respectively. HDL + rHDL-0.5 showed 12 and 7% higher FRA than that of HDL alone and HDL + rHDL-0, respectively, suggesting the synergistic effect of rHDL-containing BWA and HDL depending on the BWA dose. 

The HDL-associated paraoxonase (PON-1) assay showed that the addition of TBS or rHDL-0 (15 μg of apoA-I) resulted in similar PON activity of HDL (20 μg of protein) of 93–95 μU/L/min (Figure 10B), suggesting that addition of the rHDL-0 did not elevate the PON activity. On the other hand, the addition of BWA in rHDL resulted in higher PON-1 activity: rHDL-0.5 and rHDL-1 showed 9% and 20% higher PON activity than that of rHDL-0 in HDL, respectively. 

### 3.12. Protection of Embryo Death by BWA in rHDL

A microinjection of CML (final 800 ng) into zebrafish embryos resulted in acute embryo death with up to 22 ± 6% survivability at 24 h post-injection (Figure 11A). In contrast, the TBS-alone-injected embryo showed 73 ± 3% survivability. In the presence of CML, the co-injection of rHDL-1 resulted in the highest survivability of the injected embryo, 63 ± 3% (*p* < 0.008 versus CML+TBS), whereas rHDL-0 showed 44 ± 4% survivability (*p* < 0.032 versus CML+TBS). The co-injection of rHDL-0.5 resulted in 20% higher survivability than that of rHDL-0, approximately 53 ± 5% survivability (*p* < 0.011 versus CML+TBS), suggesting that incorporation of BWA in rHDL enhanced the protective activity from acute embryo death in a dose-dependent manner. The rHDL-1 had remarkable protective activity against CML-induced inflammatory death with the highest survivability.

The TBS-injected embryos (Figure 11B, photo a) showed a normal developmental morphology and speed, with eye pigmentation and tail elongation of more than 32 somites at 24 h post-injection (Figure 11B). In comparison, the CML + TBS-injected embryo (Figure 11B, photo b) showed severe death and developmental defects (indicated by the red arrowhead) with the weakest eye pigmentation and slowest tail elongation, less than 21 somites (indicated by the blue arrowhead). On the other hand, a co-injection of rHDL-0 (Figure 11B, photo c) improved the developmental morphology and speed with faster eye pigmentation and tail elongation (indicated by the blue arrowhead), suggesting that rHDL alone could neutralize the CML toxicity in embryos to recover the developmental speed. Interestingly, a co-injection of rHDL-0.5 (Figure 11B, photo d) and rHDL-1 (Figure 11B, photo e) resulted in the most normal developmental speed and morphology, depending on the increase in the BWA molar ratio. The CML+rHDL-1 group showed the darkest eye pigmentation and tail elongation 24 h post-injection. 

AO staining revealed an 8.7-fold increase in green fluorescence in embryos injected with CML alone, indicating a significant induction of acute apoptosis (Figure 11C,D). Conversely, co-injection of rHDL-1 resulted in a remarkable 90.5% reduction in apoptosis compared to the CML-alone group. DHE staining demonstrated a 15.8-fold rise in red fluorescence in CML-alone-injected embryos compared to the TBS-alone group, suggesting apoptosis correlated with increased ROS production. The co-injection of rHDL-1 led to a notable 98% decrease in ROS levels compared to the CML-alone group. These findings propose that incorporating BWA into rHDL enhances antioxidant and anti-inflammatory activities, ultimately promoting embryonic survivability impaired by exposure to CML.

### 3.13. BWA in rHDL Facilitated the Healing of Cutaneous Wounds 

Figure 12 presents rHDL-aided cutaneous wound healing at different time points in the presence of CML. Wound healing appeared first at 4 h post-treatment in the TBS, CML + rHDL-0, CML + rHDL-0.5, and CML + rHDL-1 treated groups. In contrast, no wound healing was observed in the CML-only-treated group. The initial wound healing with 10.6% wound closure was observed in the TBS-treated group, followed by 8.9% wound closure in the CML + rHDL-0.5 treated group at 4 h post-treatment (Figure 12B). At 6 h post-treatment, 5.7- and 4.2-fold higher wound healing was observed in the TBS and CML + rHDL-0.5 treated groups, respectively, compared to the CML-only treated group. The most noteworthy results were observed at 24 h post-treatment, where 66% wound closure was observed in the TBS-treated group. Similarly, 57% wound closure was observed in the CML + rHDL-1-treated group, which was five-fold and 1.7-fold higher than the wound healing of the CML-only and CML + rHDL-0 treated groups, respectively. The utmost wound healing, with 86.4% wound closure, was observed in the CML + rHDL-1 group, which is higher than the TBS-alone group (~82.1% wound healing), followed by 78.4% wound closure in the CML + rHDL-0.5 treated group at 48 h post-treatment. With the progression of time up to 48 h post-treatment, a maximum wound healing was observed in the CML + rHDL-1 treated groups that is significantly two-fold (*p*< 0.001) and 1.5-fold (*p* < 0.001) higher than the wound closer observed in the CML + TBS and CML + rHDL-0 treated groups, respectively. These results revealed the impact of BWA on ameliorating the rHDL wound healing activity against the CML-aggravated chronic wound. 

The histology of the skin and muscle tissue of the wounded site was evaluated by H & E staining. The results revealed the development of the epidermis (neo-epithelization) in the TBS group (Figure 12C). In contrast, a fragmented epidermis was observed in the CML and CML + rHDL-0 treated groups, indicating the role of CML in delaying wound closure. On the other hand, the CML co-treated with rHDL-0.5 and rHDL-1 groups showed remarkable epithelium development. Consequently, the wound closed. Furthermore, compact muscular tissue, as indicated by the black arrow, appeared in the TBS group (Figure 12C), indicating the recovery of the wounded site. In contrast, loosely packed muscular tissue documented the tissue injury in the CML-treated group. In contrast, the CML + rHDL-0.5 and rHDL-1-treated groups showed much more organized and compact tissue, signifying the wound-healing role of rHDL-0.5 and rHDL-1.

DHE staining suggested ROS production at the wounded site. A massive ROS production was observed in the treated group, which was significantly 2.1-fold (*p* < 0.05) higher than the ROS level in the TBS group (Figure 12C,D). Similar to the treated group, the CML + rHDL-0 treated group did not affect the inhibition of ROS production. In contrast, the wounds treated with CML + rHDL-0.5 and rHDL-1 showed a significant decrease in ROS production compared to the CML-only-treated group. The 1.5-fold (*p* < 0.05) and 1.7-fold (*p* < 0.05) lower ROS levels in the rHDL-0.5 and rHDL-1-treated groups highlight the impact of rHDL-0.5 and rHDL-1 on diminishing the ROS level and consequently the progression of wound healing.

Consistent with DHE staining, AO staining also caused the highest apoptotic cell death in CML-treated wounded tissue, which was 2.3-fold (*p* < 0.05) higher than that of the TBS-treated group (Figure 12C,D). The CML-induced apoptosis in the injured site was effectively countered by the rHDL-0.5 and rHDL-1, as evident by a significant 1.9-fold (*p* < 0.05) and 3.0-fold (*p* < 0.05) reduction in the AO fluorescence intensity. These results collectively show that rHDL-0.5 and rHDL-1 efficiently inhibited CML-induced ROS generation and subsequently inhibited apoptosis in the wounded site, leading to prompt wound recovery.

## 4. Discussion

Oxidative damage to the lipids and proteins of the cell membrane surface can induce cellular necrosis and apoptosis through chemical and biophysical changes in the lipid bilayer components via necrotic cell signaling [38]. Lipid hydroperoxide-modified protein adducts, such as *N*^ε^-(hexanoyl)lysine, were also found in oxLDL with a positive correlation with the extent of oxidation [39]. oxLDL could provoke acute cell death and foam cell formation by releasing pro-inflammatory cytokines to build up atherosclerotic plaque [40]. Therefore, inhibiting LDL oxidation is essential to prevent pro-inflammatory disease via suppressing ROS production, releasing adhesion molecules, the vicious cycle of inflammation, and apoptosis in the necrotic core while activating endothelial cells, smooth muscle cells, macrophages, and platelets [41]. On the other hand, the desirable antioxidant should protect HDL and apoA-I from oxidative modification, such as myeloperoxidase and lipoxygenase, and glycation stress, such as fructose and CML. Because HDL-associated paraoxonase activity is the principal power to exert inhibition of LDL oxidation, protection of HDL and apoA-I in the native state is essential for keeping the LDL healthy [42]. Dysfunctional HDL loses antioxidant and anti-inflammatory activity, such as paraoxonase activity; therefore, dysfunctional HDL cannot protect LDL from oxidative stress [42,43].

In general, lipid-soluble antioxidants protect cell membranes more efficiently from lipid peroxidation than water-soluble antioxidants, especially oxidized lipids in the plasma membrane and inner membrane of mitochondria. Indeed, BWA in Tween 20 showed more potent inhibitory activity against LDL oxidation than that of vit C at lower concentrations (final 10–30 μM), as shown in Figure 1. Although vit C inhibited LDL oxidation by myeloperoxidase in vitro, the dosage was too high to use practical applications, such as 50, 100, 150, and 200 mM [44]. In addition, vit C and E could not inhibit LDL oxidation by ferritin at lysosomal pH, which might help to explain why vit C and E did not reduce CVD in large clinical trials [45], which is in good agreement with the current results (Appendix A). Several phenolic compounds and flavonoids have been developed as inhibitors of LDL oxidation [46]. New pharmaceutical agents are needed to maximize the antioxidant, anti-glycation, protection of HDL, and stabilization of apoA-I via an interaction of the amphipathic helix domain. Hence, HDL functionality is more important in exerting the protective effects of HDL against cardiovascular risk than the HDL-C quantity [43]. Protection of apoA-I is critical to maintaining the antioxidant and anti-inflammatory activity of HDL because apoA-I (28 kDa) is the principal protein component in HDL particles [47]. Furthermore, apoA-I in a lipid-free state could exhibit antioxidant and anti-inflammatory activity, which can be impaired by nonenzymatic glycation [48]. 

It has been well established that CML, an advanced glycated end product, causes acute inflammation and paralysis with an increased proinflammatory cytokine, IL-6, in hepatic tissue [17,35]. The current results demonstrated that co-injection of BWA (final 100 μM) remarkably treated acute inflammation and neurotoxicity, while co-injection of vitamin C (final 100 μM) showed much weaker anti-inflammatory activity (Figure 3, Figure 4 and Figure 5). In terms of survivability and hepatic inflammation, the BWA group showed two-fold higher survivability than the vitamin C group, with 50 and 90% less infiltration of neutrophils and IL-6 production, respectively. In addition, the BWA group showed 60–70% lower serum AST and ALT levels than those of the vitamin C group. These results indicate that lipophilic antioxidant BWA exerted higher efficacy to treat CML-induced acute inflammation and neurotoxicity than hydrophilic vitamin C. However, there has been a practical limitation to using detergents such as Triton X-100 and Tween20 as vehicles, which might cause unwanted effects in dermal application and intravenous injection of BWA. Therefore, it is necessary to develop a nanoparticle to encapsulate BWA, such as rHDL, that allows it to solubilize in aqueous buffer and injectable formulations to overcome the extremely poor solubility of BWA.

This study was designed to compare the structural-functional correlations of reconstituted HDL with the increase in BWA content under the same apoA-I content and maximize the antioxidant capacity of BWA in an aqueous buffer. The apoA-I in rHDL was more stabilized as the molar ratio of apoA-I:BWA increased to approximately 1:1 (Figure 6). The rHDL particle morphology was changed to be more distinct, and the size was larger (Figure 7). Regarding functionality, the anti-glycation ability of rHDL was enhanced with more protection of apoA-I and less glycation in HDL in the presence of fructose (Figure 8). The serum fructose levels in individuals with diabetes (12.0 ± 3.8 μmol/L) were notably elevated compared to those in both healthy subjects (8.1 ± 1.0 μmol/L) and non-diabetic patients (7.7 ± 1.6 μmol/L) [49]. In addition, fructose (a ketohexose) served as a better substrate for glycation than glucose (an aldohexose), because fructose can induce eight-fold higher glucose levels in vitro. The antioxidant ability of the rHDL was strengthened as the BWA molar ratio was increased to suppress cupric ion mediated LDL oxidation (Figure 9). The FRA and PON activities of human HDL were enhanced remarkably by the rHDL as the BWA content was increased (Figure 10). The rHDL containing BWA protected against embryo death from the acute toxicity of CML by suppressing ROS production and apoptosis in a dose-dependent manner (Figure 11). BWA in rHDL facilitated cutaneous wound healing activity under CML toxicity in adult zebrafish (Figure 12). 

To the best of the authors’ knowledge, the current study is the first to show that BWA could bind with apoA-I and cholesterol to construct rHDL, suggesting that the six even-numbered long-chain aliphatic alcohols (of carbon chains 24, 26, 28, 30, 32, and 34) made a putative affinity interaction with the amphipathic helix domain of apoA-I. Hydrophobic cholesterol and triglycerides could be encapsulated in the core of HDL via interaction with endogenous apolipoproteins, such as apoA-I, which have a repeated amphipathic helix domain [50]. Using the rHDL-like nanoparticle, many non-soluble drugs can be dissolved in a physiological buffer system for systemic administration [51]. Because HDL is a major antioxidant carrier in the blood and exhibits pleiotropic activities, many synthetic formulations of rHDL have been developed for therapeutic purposes. However, there has also been a disadvantage with rHDL therapy, which is the high cost and immunological issues associated with producing large amounts of rHDL and the high dosage of intravenous injection around 40 mg/kg of body weight. To overcome this limitation, encapsulation of an efficient hydrophobic drug, such as BWA, in rHDL could be a solution to reduce the cost and dosage.

As the BWA concentration was increased, the particle size of rHDL increased up to 15% with a 1.5 nm redshift of intrinsic Trp, suggesting that the Trp in apoA-I was exposed more to the polar phase upon binding with BWA. Interestingly, this behavior of fluorophore movement differed from rHDL synthesis with policosanol (PCO) in a previous report [15]. A rHDL containing PCO (PCO-rHDL) showed a 1.9 nm blue shift of WMF, even though rHDL comprising Cuban PCO showed a 24% increase in particle diameter at a 95:5:1:1 molar ratio for POPC:FC:apoA-I:PCO. The difference between BWA and PCO is that BWA contains two odd-numbered aliphatic alcohols, C27 and C29 (1-hepacosanol and 1-nonacosanol, respectively), because policosanol is composed of eight long-chain aliphatic alcohols (of carbon chains 24, 26, 30, 32, and 34). Although the incorporation of BWA and PCO caused an increase in the particle size of rHDL to a similar extent, the movement of Trp in the amphipathic helix domain of apoA-I was different in the opposite direction. These findings indicate that Trp108 of apoA-I showed different molecular motions and fluorophore environments between BWA-rHDL and PCO-rHDL through the putative interaction of C27 and C29. The intrinsic Trp108 was more exposed to the water phase BWA-rHDL, while PCO-rHDL showed a more closed Trp108 toward the nonpolar phase of apoA-I. The BWA should have amphipathic properties between the water and lipid interfaces in the intercalated protein domain of the cell membrane to accomplish therapeutic activity for joint health benefits and gastroprotection.

On the other hand, BWA-rHDL exerted stronger anti-glycation activity and antioxidant activity than rHDL alone, with an enhancement of the FRA and PON activities in HDL. These advantageous activities could contribute to preventing the pathogenesis of atherosclerosis and diabetes via inhibition of LDL oxidation and HDL dysfunction. Although the biochemical mechanism of the antioxidant and anti-glycation activities of BWA-rHDL has not been fully elucidated, the binding of BWA and apoA-I might allow a reduction of susceptibility to oxidation and glycation stress via the movement of the hinged domain in apoA-I around Trp108. In the initial phase and propagation phase of the Maillard reaction, usually Arg and Lys are the main targets for Amadori rearrangement to yield Schiff bases. During the process of Amadori rearrangement, the oxidative function of metal ions, such as Cu^2+^, produces many carbonyl derivatives and free radicals, which can amplify oxidative stress. However, current findings could allow us to postulate that BWA in rHDL caused more stabilization of the central helices of apoA-I to avoid the initial phase of the Maillard reaction in Lys. 

Even though many papers reported that BWA exhibited potent antioxidant and anti-inflammatory activity in animal and human serum after oral supplementation, no study has identified its beneficial in vitro activity because of its insolubility in water. These enhancements of in vitro antioxidant activity of HDL by BWA-rHDL are in good agreement with previous in vivo reports that showed a decrease in various oxidized lipids and enzymes involved in inflammation, such as COX-2, by apoA-I in colon cancer and ovarian cancer [52,53]. In the same context, the consumption of BWA (5–100 mg/kg) lowered carboxyl groups (a marker of protein oxidation) and the generation of hydroxyl (*OH) radical and myeloperoxidase (MPO) activity (a marker of inflammation) and increased the catalase (CAT) activity in the gastric mucosa of rats with indomethacin ulcers [10]. Lipid peroxidation is critical to induce acute gastric mucosal injuries, pro-inflammatory cascades, and carcinogenesis [54,55].

Despite the many papers from animal and human studies showing that supplementation of BWA ameliorated gastric ulcers, esophagitis, and osteoarthritis [10,11,56], the in vitro antioxidant abilities of BWA and the mechanism of antioxidant activity after oral ingestion were not fully elucidated. Although BWA has adequate DPPH radical scavenging activity and ferric ion reduction ability in ethanol solvent assay systems [57], the current results showed that the incorporation of BWA in rHDL resulted in more enhancement of HDL-associated PON and FRA activity to inhibit LDL oxidation. The encapsulation of BWA in the rHDL is needed to maintain its antioxidant activity to suppress LDL oxidation (Figure 1), anti-glycation activity (Figure 2), and anti-inflammatory activity (Figure 3, Figure 4 and Figure 5), because an organic solvent or non-ionic detergent, e.g., Tween20, for BWA, might be toxic. The enforcement of HDL functionality by BWA encapsulation might be associated with the improvement of the barrier function by apoA-I stabilization (Figure 6 and Figure 7), reduction of glycation stress (Figure 8) and oxidative stress (Figure 9 and Figure 10), inhibition of embryonic cell apoptosis (Figure 11), and enhancement of wound healing (Figure 12). The enhancement of HDL quality and functionality by BWA was associated with that of 17-day BWA supplementation (10% in diet, wt./wt.) in ameliorating liver function and dyslipidemia to suppress interleukin-6 by increasing HDL functionality in the ethanol-induced hepatic injury zebrafish model [57]. Taken together, BWA has many strength points as a natural product, which has high bioavailability without adverse effects and exhibits potent antioxidant and anti-inflammatory activity, especially at the lipid-protein interface between cell membranes. However, BWA has extremely poor solubility in aqueous buffer, which is a weak point for developing an agent for systemic delivery to multiple organs via the bloodstream, such as gastric mucous glands and cartilaginous synovial fluid. The weakness of BWA can be overcome by encapsulation into rHDL, which allows for the development of many delivery tools into target organs to treat acute inflammatory diseases and diabetic patients with ketoacidosis or ketosis. 

## 5. Conclusions

The BWA can form discoidal rHDL via stabilization of apoA-I, which leads to enhanced binding with apoA-I and a larger particle size. The BWA-rHDL exhibited enhanced anti-glycation, antioxidant, and anti-inflammatory abilities to protect HDL/apoA-I, which can suppress oxidation of LDL, glycation of HDL, and acute embryonic cell apoptosis with promotion of wound-healing activity. The noteworthy challenge for the practical use of BWA lies in its restricted solubility in aqueous environments. Future research will focus on formulating BWA in various emulsions and encapsulating it within diverse cyclodextrins to improve both solubility and functionality. Additionally, the ratio of the BWA components would be formulated to evaluate their impact on the functionality of BWA.

## Figures and Tables

**Figure 1 antioxidants-12-02116-f001:**
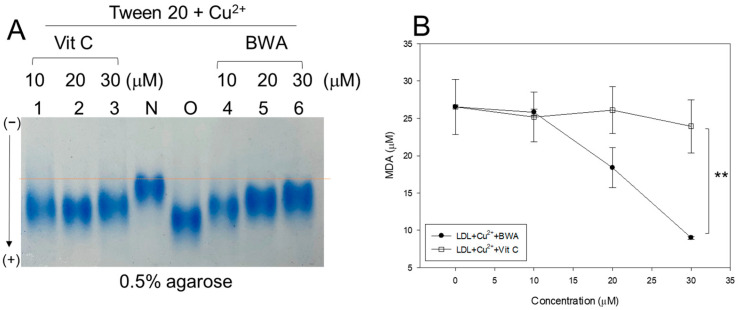
Comparison of the antioxidant ability between lipid-free BWA and vitamin C (vit-C) against cupric ion-mediated LDL oxidation. (**A**) Electromobility of LDL under a non-denatured state in the presence of Tween20 and cupric ions with BWA or vit-C after 4 h incubation at 37 °C. The red-dotted line indicates the similar electromobility of native LDL (lane N) and BWA-treated ox-LDL (lane 6). (**B**) Quantification of malondialdehyde in the LDL by thiobarbituric acid reactive substances (TBARS) assay. **, *p* < 0.01.

**Figure 2 antioxidants-12-02116-f002:**
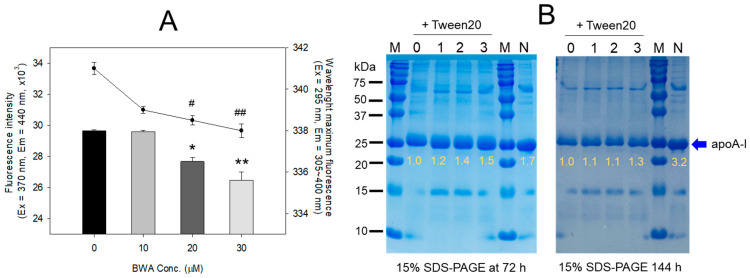
Anti-glycation ability of lipid-free BWA in Tween20 against CML-mediated glycation. (**A**) Fluorescence intensity (Ex = 370 nm, Em = 440 nm) and wavelength maximum fluorescence (Ex = 295 nm, Em = 305–400 nm) of HDL in the presence of CML (final 400 μM) and BWA (final 10, 20, and 30 μM) at 144 h incubation. *, *p* < 0.05 versus HDL + CML; **, *p* < 0.05 versus HDL + CML; ^#^, *p* < 0.05 versus HDL + CML; ^##^, *p* < 0.05 versus HDL + CML. (**B**) Electrophoretic patterns of HDL in the presence of CML (final 400 μM) and BWA (final 10, 20, and 30 μM) in Tween20 (final 30%) at 72 h and 144 h incubation (15% SDS-PAGE). The protein bands were visualized by Coomassie blue staining. The yellow font indicates the band intensity of apoA-I from three distinct SDS-PAGEs. Lane 0, CML + HDL; lane 1, CML + HDL + BWA (final 10 μM); lane 2, CML + HDL + BWA (final 20 μM); lane 3, CML + HDL + BWA (final 30 μM); Lane N, native HDL; lane M, molecular weight marker.

**Figure 3 antioxidants-12-02116-f003:**
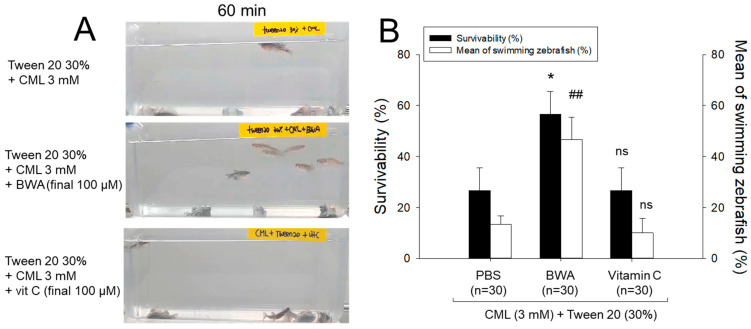
Swimming ability and survivability after 60 min post-injection of carboxymethyllysine (CML) and Tween20 (final 30%) with either BWA (final 100 μM) or Vit-C (final 100 μM). (**A**) Static representation of the swimming behavior of zebrafish after 60 min post-injection of carboxymethyllysine (CML) (250 μg, final 3 mM) and Tween 20 (final 30%) with either BWA or Vit C. (**B**) Survivability and percentage of swimming zebrafish after 60 min post-injection of CML (250 μg, final 3 mM) and Tween 20 (final 30%) with either BWA and Vit C. *, *p* < 0.05 versus PBS for survivability; ^##^, *p* < 0.01 versus PBS for swimming; ns, not significant.

**Figure 4 antioxidants-12-02116-f004:**
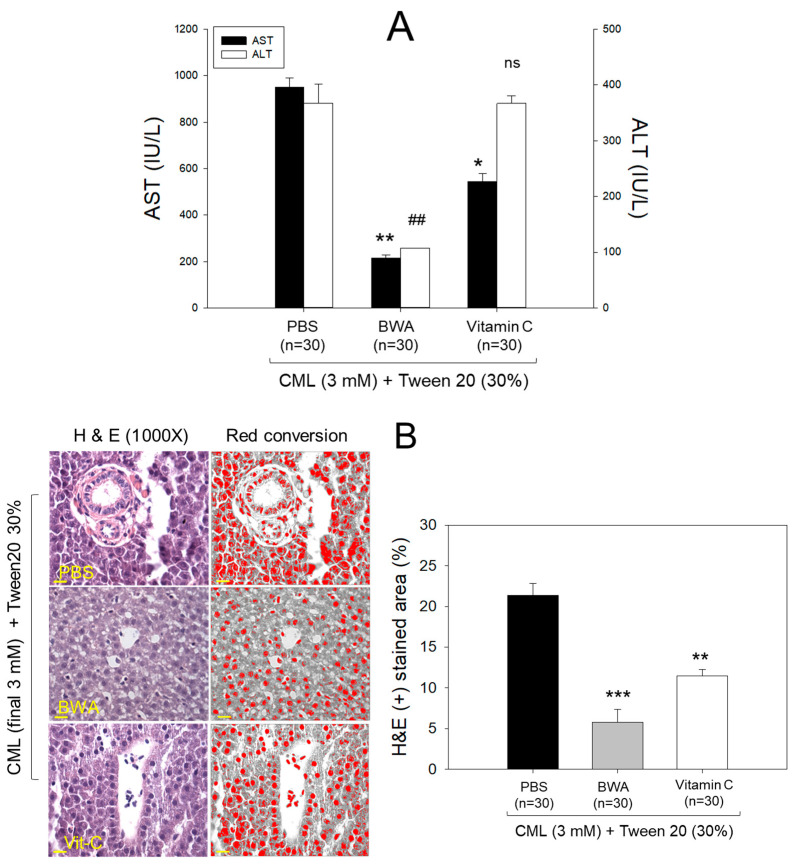
BWA treatment mitigates carboxymethyllysine (CML)-induced hepatic damage. (**A**) Hepatic enzyme levels in zebrafish plasma were quantified after 60 min post-injection of different treatments. AST (aspartate aminotransferase) and ALT (alanine aminotransferase) levels were assessed, with significant differences denoted as follows: *, *p* < 0.05; **, *p* < 0.01 for AST level compared with PBS group; ^##^, *p* < 0.01 for ALT level compared to PBS groups; ns represents nonsignificant difference. (**B**) A hepatic examination was conducted on zebrafish hepatic tissue following carboxymethyllysine injections (CML) along with BWA or Vit-C. The images depict neutrophil infiltration through Hematoxylin and eosin (H&E) staining and subsequent conversion to red intensity using Image J software (Scale bar = 10 μm). Statistical analysis indicates **, *p* < 0.01 and ***, *p*<0.001 compared to PBS.

**Figure 5 antioxidants-12-02116-f005:**
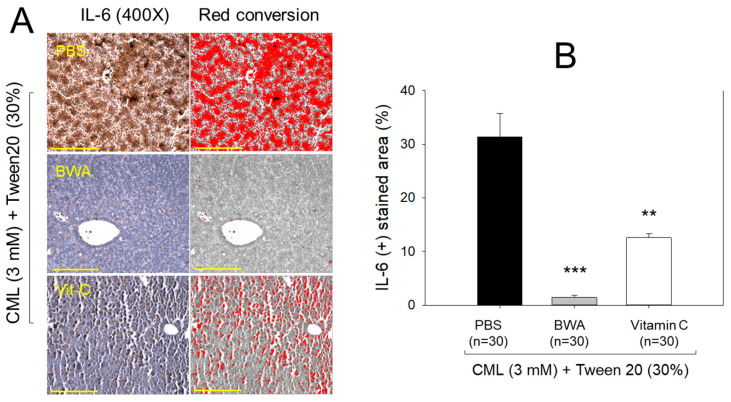
Quantification of the interleukin (IL)-6-stained area in hepatic tissue from immunohistochemistry (IHC) with carboxymethyllysine (CML)-injected zebrafish. (**A**) Photographs show representative images of the IHC using IL-6 antibody-stained hepatic tissue from each group. The yellow scale bar indicates 100 μm. (**B**) Graphical comparison of the IL-6 antibody-stained area with the red conversion area from brown intensity using Image J software. IL-6, interleukin-6; BWA, beeswax alcohol. **, *p* < 0.01 versus PBS control; ***, *p* < 0.001 versus PBS control.

**Figure 6 antioxidants-12-02116-f006:**
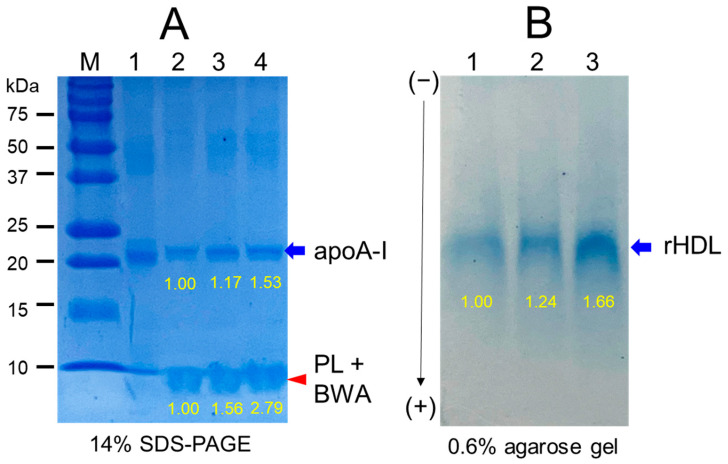
Electrophoretic patterns of recombinant high-density lipoproteins (rHDL) containing BWA are presented in both denatured and non-denatured states. (**A**) Electrophoretic profiles of each rHDL were observed under denatured conditions using 14% SDS-PAGE with 9 μg of protein/lane. A red arrowhead indicates phospholipid and BWA debris. Coomassie brilliant blue staining (final 1.25%) was employed to visualize apoA-I and phospholipid. The lanes include M, molecular weight standard; lane 1, lipid-free apoA-I alone; lane 2, rHDL-0; lane 3, rHDL-0.5; lane 4, rHDL-1. (**B**) Native electrophoresis of each rHDL was conducted under non-denatured conditions using 0.6% agarose with 10 μg protein/lane. This analysis aimed to compare electromobility based on the three-dimensional structure of apoA-I/HDL and its oxidation extent. The visualization of apoA-I in rHDL was achieved through Coomassie brilliant blue staining (final 1.25%). The lanes comprise: lane 1, rHDL-0; lane 2, rHDL-0.5; lane 3, rHDL-1.

**Figure 7 antioxidants-12-02116-f007:**
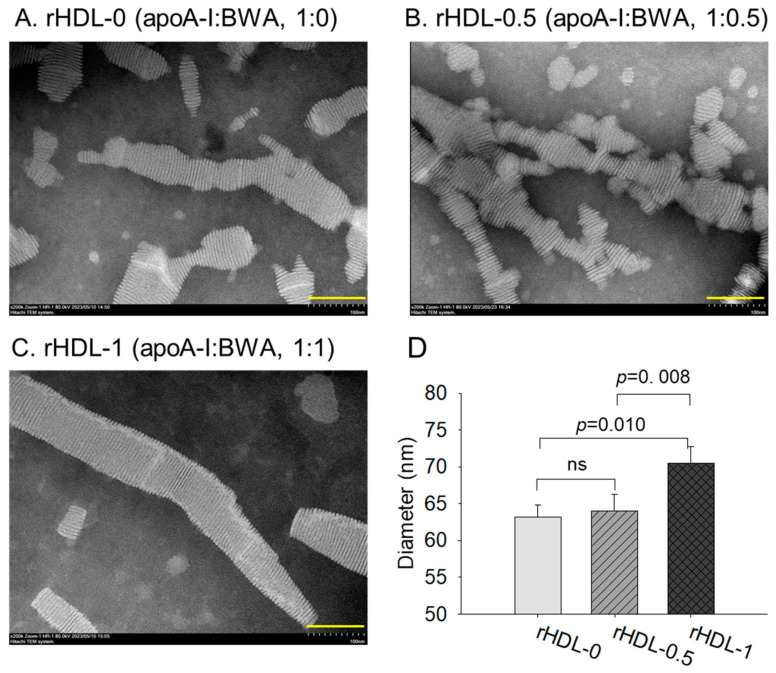
TEM image analysis of each rHDL containing BWA with 40,000× magnification. (**A**) rHDL-0. (**B**) rHDL-0.5. (**C**) rHDL-1. Among rHDLs, rHDL-1 (apoA-I:BWA, 1:1) had the largest size, as shown in the inset graph (**D**). The rHDL-1 exhibited the most distinct and uniform discoidal particle shape with a rouleaux pattern (scale bar: 100 nm).

**Figure 8 antioxidants-12-02116-f008:**
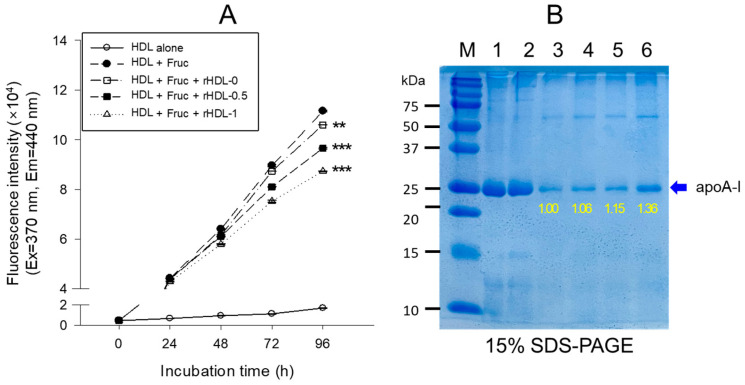
The anti-glycation effect of rHDL-comprising BWA against fructose-treated HDL was assessed through fluorescence spectroscopic analysis and electrophoretic patterns. (**A**) For fluorescence spectroscopy (Ex = 370 nm, Em = 440 nm), HDL (1 mg/mL of protein) was co-treated with fructose (final concentration of 250 mM) and each rHDL (0.2 mg/mL of apoA-I) containing BWA (final concentration of 1.3 and 2.6 μg/mL of rHDL-0.5 and rHDL-1, respectively) during a 96 h incubation period. ** *p* < 0.01 versus HDL + Fruc; *** *p* < 0.001 versus HDL + Fruc. (**B**) Electrophoretic analysis (15% SDS-PAGE) of HDL (5 μg/lane) treated with fructose + different rHDL. The protein bands were visualized by Coomassie brilliant blue staining, and the molecular weight standards (Bio-Rad Cat # 161-0374) were used for reference. Samples in different lanes are lane 1, HDL alone (0 h); lane 2, HDL alone (72 h); lane 3, HDL + Fruc; lane 4, HDL + Fruc + rHDL-0; lane 5, HDL + Fruc + rHDL-0.5; and lane 6, HDL + Fruc + rHDL-1.

**Figure 9 antioxidants-12-02116-f009:**
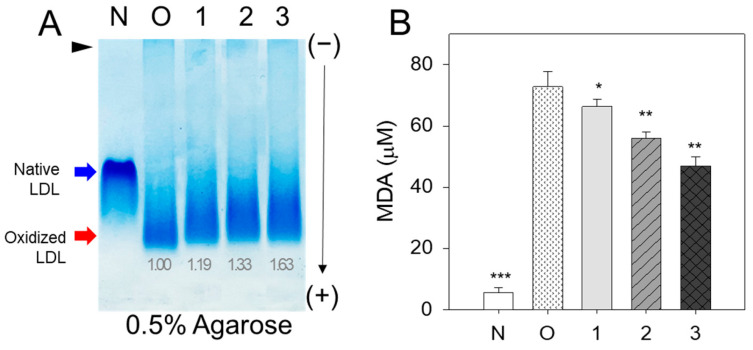
Assessment of the antioxidant capabilities of rHDL-enriched BWA against LDL oxidation provoked by cupric ions. (**A**) Evaluation of the relative electromobility of a mixture containing LDL (15 μg of protein) with rHDL (6 μg of protein) in a non-denatures state on a 0.5% agarose gel. Apo-B in LDL was visualized through Coomassie Brilliant Blue staining. (**B**) Assessment of oxidation extent via TBARS assay using malondialdehyde (MDA) as the standard. Each rHDL treatment was statistically compared with LDL+CuSO_4_ (oxLDL) using an independent *t*-test. *, *p* < 0.05 versus oxLDL alone; **, *p* < 0.01 versus oxLDL alone; ***, *p* < 0.001 versus oxLDL alone. lane N, native LDL; lane O, oxLDL; lane 1, LDL + CuSO_4_ + rHDL-0; lane 2, LDL + CuSO_4_ + rHDL-0.5; lane 3, LDL + CuSO_4_ + rHDL-1.

**Figure 10 antioxidants-12-02116-f010:**
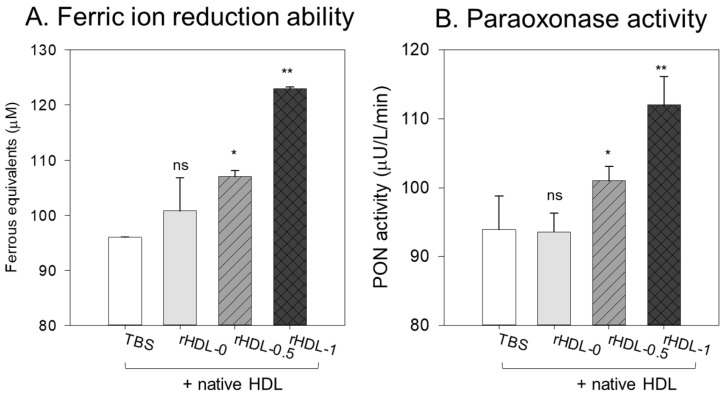
Determination of the antioxidant potential of BWA-supplemented rHDL. (**A**) Ferric ion reduction (FRA) ability as the equivalent concentration of vitamin C (μM), representing the reduction of ferric ions (μM)/h. (**B**) Paraoxonase (PON)-1 activity is expressed as the initial velocity of p-nitrophenol production per minute (μU/L/min) during the 180 min incubation period. *, *p* < 0.05; **, *p* < 0.01 compared to TBS + HDL alone; ns represents the non-significant difference.

**Figure 11 antioxidants-12-02116-f011:**
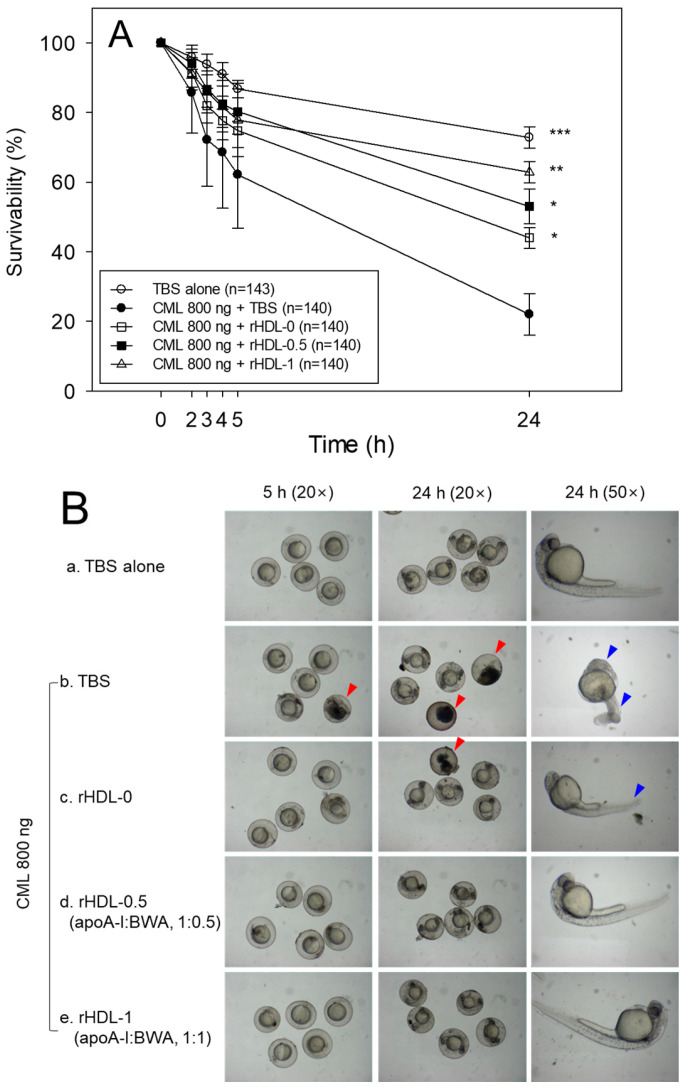
Comparison of the viability and embryonic development between reconstituted high-density lipoproteins (rHDLs) containing BWA in the presence of carboxymethyllysine (CML, 800 ng). (**A**) Embryonic survivability within 24 h post-injection of each rHDL and CML from three independent experiments. Statistical significance is indicated as follows: *, *p* < 0.05; **, *p* < 0.01; ***, *p* < 0.001 versus CML alone. An independent *t*-test was utilized for statistical comparison among multiple groups. (**B**) Morphological assessment of embryonic development at 5 and 24 h post-injection. Defects and embryo mortality are highlighted by the red arrow. The slowest developmental progress in eye pigmentation and tail elongation is indicated by the blue arrow. (**C**) Fluorescence imaging of acridine orange (AO, Ex = 505 nm, Em = 535 nm) and dihydroethidium (DHE, Ex = 585 nm, Em = 615 nm) stained embryos at 5 h post-injection (Scale bar = 250 μm). a. TBS-alone group; b. CML + TBS injected group, while c, d, and e are CML + rHDL or rHDL-BWA (1:0.5) or CML + rHDL-BWA (1:1) injected groups, respectively. (**D**) Quantifying fluorescence intensity from AO and DHE-stained embryos using Image J software. Statistical differences among multiple groups were assessed using an independent *t*-test. *, *p* < 0.05; ***, *p* < 0.001 represent the statistical difference for the AO-stained area between the groups compared to CML+TBS. ^#^, *p* < 0.05; ^##^, *p* < 0.01; ^###^, *p* < 0.001 represent the statistical difference for the DHE-stained area between the groups compared to CML + TBS.

**Figure 12 antioxidants-12-02116-f012:**
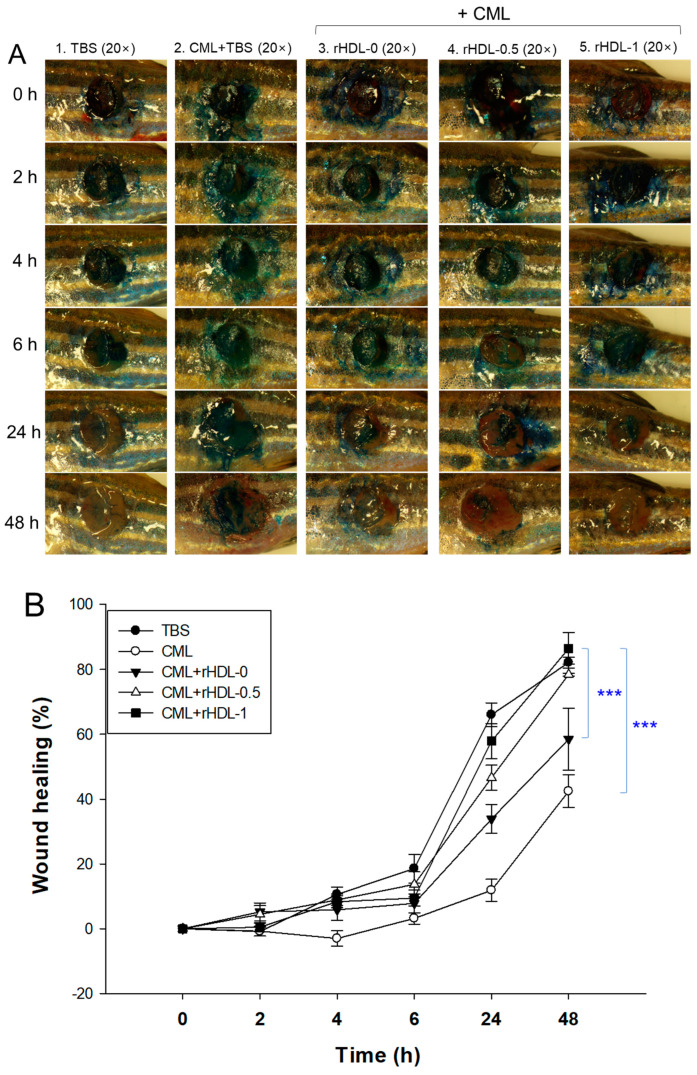
Comparative wound healing effect of rHDL composed of different proportions of beeswax alcohol (BWA) against carboxymethyllysine (CML) impaired cutaneous wounds in zebrafish. (**A**) Pictorial view of the wounded area stained with methylene blue (0.1% (*w*/*v*) final). (**B**) A time-dependent wound healing up to 48 h post-treatment. The wound healing was determined by contrasting the stained wound area measured at various time intervals against the wound-stained area at 0 h. The *p*-value reported the pairwise statistical differences obtained from ANOVA, using a Tukey test for post hoc analysis. ***, *p* < 0.001. (**C**) Hematoxylin and eosin (H& E), dihydroethidium (DHE), and acridine orange (AO) staining of the wounded tissue at 48 h post-treatment. The black and red arrows indicate compact and loosely packed muscular tissue, respectively (scale bar = 100 μm). (**D**) Dihydroethidium (DHE) and acridine orange (AO) staining of the wounded tissue at 48 h post-treatment. The fluorescent intensity in DHE and AO staining was measured using Image J software. The letters in italic font (*a–d*, *A–D*) above the bar charts indicate the significant statistical differences (*p* < 0.05) among the groups.

**Table 1 antioxidants-12-02116-t001:** Characterization of rHDL containing different ratios of beeswax alcohol.

Name	Initial Molar Ratio POPC:FC:apoA-I:BWA ^1^	Final Amount (μg) in 0.7 mL (POPC:FC:apoA-I:BWA)	WMF (nm)	Diameter (nm)
rHDL-0	95:5:1:0	2600:70:1000:0	330.9 ± 0.1	61.2 ± 1.6
rHDL-0.5	95:5:1:0.5	2600:70:1000:7.5	331.0 ± 0.0	65.4 ± 2.3
rHDL-1	95:5:1:1	2600:70:1000:15	331.4 ± 0.1	70.5 ± 2.3

apoA-I, Apolipoprotein A-I (MW = 28,000); BWA, beeswax alcohol (MW = 429.1); FC, free cholesterol (MW = 386.7); POPC, palmitoyloleoyl phosphatidylcholine (MW = 760.1); WMF, wavelength maximum fluorescence; MW, molecular weight. ^1^ Total amount of long-chain aliphatic alcohols in BWA: 859.79 mg/g, 1-tetracosanol (C24): 63.67 mg/g, 1-hexacosanol (C26): 114.28 mg/g, 1-octacosanol (C28): 135.96 mg/g, 1-triacotanol (C30): 292.25 mg/g, 1-dotriacotanol: 222.97 mg/g, residual aliphatic alcohol: 30.62 mg/g.

## Data Availability

The data used to support the findings of this study are available from the corresponding author upon reasonable request.

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
