# Peer review of "Enhancement of Antioxidant and Anti-Glycation Properties of Beeswax Alcohol in Reconstituted High-Density Lipoprotein: Safeguarding against Carboxymethyllysine Toxicity in Zebrafish"

_antioxidants, 2023, doi:10.3390/antiox12122116_

Round 1

Reviewer 1 Report (Previous Reviewer 1)

Comments and Suggestions for Authors

The authors addressed most of the required comments and changed the manuscript accordingly. The manuscript sounds now better and now warrants publication in Antioxidants journal.

Comments on the Quality of English Language

fine

Author Response

Thank you very much for your appreciation and acceptance.

Reviewer 2 Report (New Reviewer)

Comments and Suggestions for Authors

In this study, the authors investigated the antioxidant and antiglycation activities of alcoholic beeswax extract ( BWA) in rHDL and compared the effects of BWA with vitamin C against carboxymethyllysine toxicity in zebrafish. The study complements findings from animal and clinical studies showing  antioxidant activities of BWA on lipoperoxidation, protein oxidation and others.

A somewhat complicated study is well and carefully written and well documented. I only have a few comments:

Comparing the antioxidant effects of BWA with those of vitamin C is incorrect. BWA is a lipophilic substance and can act in the prevention of the initial stages of lipoperoxidation. In contrast, hydrophilic vitamin C acts in the regeneration of oxidized and reduced glutathione. Therefore, vitamin C cannot affect the initial phases of lipoperoxidation (measured as malondialdehyde in the study), which take place in a lipophilic environment.

The correct analysis of LDL oxidizability is according to the formation of conjugated dienes measured spectrometrically at 235 nm every 3 minutes after adding Cu2+ to the medium. The detected lag phase before the formation of dienes is an indicator of oxidation resistance, i.e. the presence of substances with antioxidant properties. The formation of MDA can be influenced by other factors in the oxidation chain and is therefore not an indicator of the antioxidant protection of LDL.

BWA contains a mixture of six aliphatic alcohols but its effectiveness is compared to PBS. Why did the authors not compare the effects of BWA with low alcohol concentration?

Beeswax is a highly efficient accumulator of heavy metals such as Fe, Cr, Zn, Cu, Ni, Mn, Pb, Cd and Co. What problem can this fact affect the activity of BWA?

Author Response

Thank you for your valuable comments and suggestions. 

Please find attached doc as point-to-point response.

Reviewer 3 Report (New Reviewer)

Comments and Suggestions for Authors

The manuscript entitle: “Antioxidant and anti-glycation abilities of beeswax alcohol  were more enhanced in reconstituted high-density lipoprotein  with potent safeguarding and protection against acute toxicity  of carboxymethyllysine in zebrafish adult and embryo”, the antioxidant and antiglycation capacities of beeswax alcohol on high-density lipoprotein in zebrafish adults and embryos are analyzed, showing that BWA-rHDL has improved antiglycation, antioxidant and anti-inflammatory properties to protect HDL/apoA-I, decreasing oxidation of LDL, glycation of HDL and acute apoptosis of embryonic cells with promotion of wound healing activity.

The topic is very interesting and can be considered a further step in the studies carried out by the research team in previous works. The manuscript contains all the complete sections, with adequate experiments to be able to draw interesting conclusions, however their development requires major revision for publication.

 Below are aspects to improve:

First: the title should be shortened, it is almost an introduction. The title should summarize the main idea of the manuscript in a simple way and not develop it almost completely.

Second: the summary is also too long, mixing objectives, conclusions and materials and methods. It is necessary for the authors to organize the important sections and shorten the content.

Third: The text has 41% coincidences, omitting those from the preprint are 23% corresponding to the previous work: https://doi.org/10.3390/ijms24043186, distributed in the Introduction, Material and Methods, Results, including the captions of figures and finally the conclusion. Therefore, authors should avoid this large number of coincidences by carefully reviewing the manuscript after running it through an anti-plagiarism program.

Fourth: The limitations of the study are not indicated in the manuscript. Addressing limitations helps provide a balanced perspective and suggests avenues for future research. Please review and complete the discussion and conclusions, taking this aspect into account.

Comments on the Quality of English Language

Minor editing of English language required

Author Response

Thank you for your valuable comments and suggestions. 

Please find attached doc as point-to-point response.

Round 2

Reviewer 3 Report (New Reviewer)

Comments and Suggestions for Authors

The authors have made the changes proposed by the reviewer, so the paper is accepted in its present form.

Comments on the Quality of English Language

Minor editing of English language required

This manuscript is a resubmission of an earlier submission. The following is a list of the peer review reports and author responses from that submission.

Round 1

Reviewer 1 Report

Comments and Suggestions for Authors

The manuscript has many problems. The methods are not clearly presented, and it is difficult to follow the results.

 Authors are asked to revise many sentences and present data in a better way.

The manuscript appears to be very unclear.

 2.2. Purification of human lipoproteins--this section lacks a lot of information, e.g., in line 127 enter the initial volume of serum and define the parameters of the final yield.

 2.3. Purification of human apoA-I the method must be clearly described to understand the procedure used. also. the rHDL must be quantified to know starting doses and final yields.

 3.6. Anti-glycation activity of BWA in rHDL with fructose (final 250 mM) why not glucose?

The authors need to explain and convince the reader why they used CML and fructose as glycating agents.

It would have been more appropriate to use glucose.

Fructose and CML are not only the main glycating agents in vivo. In fact, although fructose is 10 times more reactive than glucose in protein glycation, its plasma concentration is only 1% of that of glucose.

 The SDS-PAGE of figure 2 and 3 are not well described also in the methods.

 Figure 4 TEM missing size bar

 It is not clear why both agarose electrophoresis and SDS-PAGE are used. Methods and also the purpose of the two different electrophoretic techniques should be described.  

PON-1 activity line 432 . in the text and section 3.8 is mentioned PON-1 activity but is not found in the methods. In addition, the significance of this activity must be specified.

The authors should explain why blood products such as HDL should be used to enhance the role of BWA. There are many clinical and economic problems and disadvantages for this type of application with blood products.

Comments on the Quality of English Language

fine

Author Response

(The authors gave the same response as above.)

Reviewer 2 Report

Comments and Suggestions for Authors

The paper of Cho et al addresses the use of beeswax alcoholic extract as an antioxidant, anti-protein glycation and wound healing agent in zebrafish embryos. The idea is very interesting and as far as I know, the most interesting aspect is the anti- protein glycation effector this extract.

But beyond this, the paper has not the level for the publication in Antioxidant journal.

Firstly, in Materials and Methods section:

1. a more detailed description of the synthesis of reconstituted HDL is absolutely necessary

2. For protein determination, the authors must specify what type of ultracentrifugation have used, and what type of method: Lowry, Bradford, or both.

The Discussion section does not contain the explanation of the biochemical mechanism  responsible for the increased anti-oxidant and anti-protein glycation effects generated by the increased level of BWA in rHDL. 

Author Response

(The authors gave the same response as above.)

Reviewer 3 Report

Comments and Suggestions for Authors

This paper focuses on the antioxidant and anti-glycosylation properties of beeswax alcohol in recombinant high-density lipoproteins in recombinant HDL, whether it has enhanced antioxidant and anti-glycosylation properties, whether it has effective wound healing activity and protects zebrafish embryos from the acute toxicity of carboxymethyl lysine acute toxicity of carboxymethyl lysine in zebrafish embryos. However, there are some deficiencies in the manuscript of this thesis, and further modifications are suggested as follows. The specific modifications are as follows:

1.     Please include the full name of HDL in the abstract, and the full name is required for the first appearance.

2.     Please explain what is meant by MFDS in parentheses in the second paragraph of the introduction, and check whether there are any problems.

3.     Increase keyword low-density lipopro-teins (LDL).

4.     Figure 2, labeled, the red font is not present in figure B. The author is requested to double-check and reinterpret.

5.     Please explain result 3.1 the quantification of malondialdehyde (MDA) in the LDL mixture showed that the BWA treatment caused a significant decrease in MDA in a dose-dependent manner up to 32%, and how the percentages are referred to in the text several times in the context of comparative analyses of the experimental results, so please double-check the results and explain them one by one.

6.     The staining, fluorescence, and electron microscopy images in the text are not to scale.

7.     Fig., figure, photo, graph Please standardize the format of the full text.

8.     Please add how BWA, which is not readily soluble in water, is encapsulated in HDL. please explain the concentration of the liquid encapsulated in HDL.

9.     In the notes to figure 9, p is not italicized, please check the statistically significant p-values written throughout the text.

10.  In the discussion section of the paper, please specify the strengths and weaknesses of beeswax alcohol, the implications of the research and the prospects for its development.

11.  Please harmonize the format of references.

Author Response

(The authors gave the same response as above.)

Round 2

Reviewer 1 Report

Comments and Suggestions for Authors

The manuscript has been improved but there still remain many unclear points in the methods and working choices of the Authors.

In my opinion, I believe that the current version of the manuscript, even if revised, has not achieved  the requirements for publication in Antioxidants journal

Comments on the Quality of English Language

good quality

Reviewer 2 Report

Comments and Suggestions for Authors

In this form the paper can be accepted for publishing in Antioxidant journal.

Reviewer 3 Report

Comments and Suggestions for Authors

After revision, the content of this article is substantial, the structure is reasonable, and the experimental evidence is sufficient. It is recommended to publish it.